# Diagnostic Approach for Accurate Diagnosis of COVID-19 Employing Deep Learning and Transfer Learning Techniques through Chest X-ray Images Clinical Data in E-Healthcare

**DOI:** 10.3390/s21248219

**Published:** 2021-12-09

**Authors:** Amin Ul Haq, Jian Ping Li, Sultan Ahmad, Shakir Khan, Mohammed Ali Alshara, Reemiah Muneer Alotaibi

**Affiliations:** 1School of Computer Science and Engineering, University of Electronic Science and Technology of China, Chengdu 611731, China; khan.amin50@yahoo.com; 2Department of Computer Science, College of Computer Engineering and Sciences, Prince Sattam Bin Abdulaziz University, Alkharj 11942, Saudi Arabia; s.alisher@psau.edu.sa; 3College of Computer and Information Sciences, Imam Mohammad Ibn Saud Islamic University (IMSIU), Riyadh 11432, Saudi Arabia; mamalsharaa@imamu.edu.sa (M.A.A.); RMALotaibi@imamu.edu.sa (R.M.A.)

**Keywords:** COVID-19 diagnosis, accuracy, transfer learning, convolution neural network, multi classification, clinical images data

## Abstract

COVID-19 is a transferable disease that is also a leading cause of death for a large number of people worldwide. This disease, caused by SARS-CoV-2, spreads very rapidly and quickly affects the respiratory system of the human being. Therefore, it is necessary to diagnosis this disease at the early stage for proper treatment, recovery, and controlling the spread. The automatic diagnosis system is significantly necessary for COVID-19 detection. To diagnose COVID-19 from chest X-ray images, employing artificial intelligence techniques based methods are more effective and could correctly diagnosis it. The existing diagnosis methods of COVID-19 have the problem of lack of accuracy to diagnosis. To handle this problem we have proposed an efficient and accurate diagnosis model for COVID-19. In the proposed method, a two-dimensional Convolutional Neural Network (2DCNN) is designed for COVID-19 recognition employing chest X-ray images. Transfer learning (TL) pre-trained ResNet-50 model weight is transferred to the 2DCNN model to enhanced the training process of the 2DCNN model and fine-tuning with chest X-ray images data for final multi-classification to diagnose COVID-19. In addition, the data augmentation technique transformation (rotation) is used to increase the data set size for effective training of the R2DCNNMC model. The experimental results demonstrated that the proposed (R2DCNNMC) model obtained high accuracy and obtained 98.12% classification accuracy on CRD data set, and 99.45% classification accuracy on CXI data set as compared to baseline methods. This approach has a high performance and could be used for COVID-19 diagnosis in E-Healthcare systems.

## 1. Introduction

COVID-19 is a transferable illness that is caused by the severe acute respiratory syndrome coronavirus 2 (SARS-CoV-2) [1]. COVID-19 is very quickly spread, and numerous people have suffered and died from this global pandemic. The efficient and accurate identification of COVID-19 is a big challenge to researchers and medical experts. Effective diagnosis technologies are significantly necessary for effective treatment and recovery of COVID-19 at an early stage. The Coronaviruses are a big family of viruses and SARS-CoV-2 is a ribonucleic acid (RNA) virus that belongs to coronaviruses. The COVID-19 can be diagnosed through different methods such as medical symptoms (fever, cough, dyspnea, and pneumonia), epidemiological history, positive pathogenic testing, positive chest X-ray, and CT images. However, two virus detection methods are used: detection through nucleic acids of the virus RNA or through antibodies generated in the patient’s immune system [1]. Thus, the diagnosis of COVID-19, clinical imagining such as chest X-ray, computer tomography (CT), and real-time polymerase chain reaction (RT-PCR) are suitable for accurate and efficient detection. Furthermore, chest CT scan images are employed to test the severity of lung involvement of COVID-19 positive subjects and provide depth information to analyze the pathogenesis of the disease [2].

Artificial intelligence (AI) techniques and their application are widely used in different domains, particularly computer vision and imaging. The diagnosis of disease employed artificial intelligence techniques on clinical images data has great applications. Medical images data such as X-ray and CT scans are mostly analyzed by applied AI techniques to diagnose diseases such as COVID-19. Due to AI these diseases are effectively diagnosis at early stages and ensuring the proper treatment and recovery of patients. The AI-based computer-aided diagnosis (CAD) systems accurately diagnose diseases than medical professionals because the medical experts do not correctly interpret the images of chest X-ray and CT Scan to diagnosis the disease at an early stage [3,4,5,6,7,8].

To detect the disease, various non-invasive-based methods have been proposed employing different kinds of images data such as X-rays [9,10,11], CT scans [12,13,14,15] and both X-rays and CT scans [16]. In these non-invasive-based techniques, mostly Machine Learning (ML) and Deep Learning (DL) techniques are employed to diagnose disease. The diagnosis of disease from images data using a convolutional neural network (CNN) model has gained very high popularity, and mostly a CNN classifier is used for classification and analysis of medical images data [17]. The CNN model has the capability to extract more related features from data for correct images classification [18]. The CNN model needs more inputs data for training of the model, however, this problem can be tackled by incorporating data augmentation [19] and transfer learning techniques [20].

In literature, various methods have been proposed for COVID-19 diagnosis using ML and DL approaches by researchers. In all these proposed methods, X-rays and CT scans images data have been used in AI algorithms to diagnose COVID-19. Numerous COVID-19 diagnosis AI-based CAD systems developed for quick and accurate detection to assist the E-healthcare systems in the world to handle this critical pandemic [21,22,23,24,25]. In these proposed models, mostly CNN and other CNN architectures, transfer learning, and data augmentation have been used to diagnosis COVID-19. Due to the lack of more data for training of the model, data augmentation techniques have been applied on X-rays and CT scans images data to increased data size [1].

Song et al. [26] designed a system for detection of COVID-19 and incorporated a detailed relation extraction neural network (DRE-Net) architecture which is named Deep Pneumonia. They trained the proposed model using a CT images data set. The data consist of 88 COVID-19 patient subjects, 101 bacteria pneumonia patient subjects, and 86 healthy subjects. The proposed model evaluated and obtained 86% and 95% accuracy and area under the curve (AUC), respectively. Wang et al. [23] proposed a COVID-19 diagnosis method employing deep learning algorithms and CT scan images. They extract features from CT scan images and then used these extract features for the classification of COVID-19 images from the viral pneumonia images. The data set used has 1065 CT images with 70% viral pneumonia and 30% COVID-19 images and the proposed method achieved 89.5% classification accuracy.

Xu et al. [24] proposed an integrated system based on CNN and ResNet models for COVID-19 diagnosis using CT scan images data and obtained 86.7% accuracy of classification. Chowdhury et al. [27] proposed a COVID-19 diagnosis method employing deep learning techniques and chest X-ray images data. The proposed method obtained classification 97.9%. Tawsifur et al. [28] proposed a diagnosis method for COVID-19 detection using chest X-ray images. They employed deep learning techniques to diagnosis the COVID-19. The proposed method achieved 95.11% accuracy.

Loddo et al. [29] proposed a COVID-19 diagnosis method employing CNN different architectures for accurate detection of COVID-19. In the proposed method development, two CT scan images data sets COVIDx CT-2A and COVID-CT have incorporated for evaluation of proposed model. The proposed method has been evaluated using different evaluation metrics and in terms of accuracy among the other CNN architectures the VGG19 obtained 98.87% on COVIDx CT-2A data set.

Gunraj et al. [30] proposed improved deep learning based diagnosis system (COVIDNet CT-2) for COVID-19 identification using CT scan images clinical data. The proposed method has been evaluated using different evaluation metrics and in terms of accuracy the method achieved 98.1% accuracy. Hu et al. [31] proposed a COVID-19 identification method using weakly supervised deep learning strategy and evaluated the proposed method using chest CT scan images data. The performance of proposed method achieved high predictive performance.

Khalifa et al. [32] proposed a COVID-19 diagnosis method using Generative adversarial networks (GAN) with a fine-tuned deep transfer learning. The proposed method has been evaluated using chest X-ray images data. They used 10% of data from data set for training and generate 90% data for training using GAN proposed model. Different transfer learning models such as Resnet18, Squeeznet, GoogLeNet, and AlexNet are used for detection of pneumonia. Furthermore, different performance evaluation metrics were used for model evaluation, but in terms of accuracy the proposed method obtained 99% accuracy. Wang et al. [33] proposed a deep convolutional Neural Network for the diagnosis of COVID-19 using data from chest X-ray images. The proposed model achieved 93.3 percent accuracy.

In this research paper, we have proposed a (R2DCNNMC) model for the diagnosis of COVID-19. In the designing of the method, we have incorporated a deep learning two-dimensional Convolution Neural Networks (2DCNN) model for extraction of deep features from chest X-ray images data and used these for final classification. In addition, transfer learning, and data augmentation techniques have been employed to increase the training process of the 2DCNN model. Furthermore, we have used the hold-out cross-validation technique for hyperparameters tuning and best model selection. The performance evaluation metrics have been computed for model performance evaluation. The performance of the baseline methods in terms of accuracy is compared with the proposed R2DCNNMC model.

The innovations of this study are summarized as follows:A deep learning-based R2DCNNMC model is proposed for detection of COVID-19 employed chest X-ray images data.To improve the predictive performance of the 2DCNN model we have used transfer learning and data augmentation techniques to improve the training process for effective training of the 2DCNN model.The proposed 2DCNNMC model performances have been evaluated by using various performance evaluation metrics.The proposed 2DCNNMC model obtained high performance compared to baseline models.

The remaining manuscript is arranged as follows: In Section 2, the data sets used in the work and proposed method methodology are discussed. Experiments are carried out and discussed in Section 3. Conclusions and future work are reported in Section 4.

## 2. Materials and Method

### 2.1. Data Collection

In this study, two data sets are used for the evaluation our model. The COVID-19-Radiography-Dataset (CRD) is a database of chest X-ray images for COVID-19 positive cases along with Normal, Viral Pneumonia images, and Lung Opacity. The data set has included 3616 COVID-19 positive cases, 10,192 Normal, 6012 Lung Opacity, and 1345 Viral Pneumonia images. The OVID-19-Radiography-Dataset is available on the Kaggle machine learning repository (https://www.kaggle.com/tawsifurrahman/COVID-19-Radiography-Database (accessed on 10 March 2021)). The second one chest X-ray (COVID-19, Pneumonia) (CXI) data set is achieved from Kaggle repository (https://www.kaggle.com/prashant268/chest-X-ray-COVID-19-pneumonia (accessed on 10 March 2021)). The data set has 6432 chest X-ray images, which belong to three classes (COVID-19, Normal, PNEUMONIA). The data set contain 576 COVID19, 1583 Normal, and 4273 PNEUMONIA images, respectively.

### 2.2. Proposed Method Background

The method background described in the below subsection in detail.

#### 2.2.1. Convolutional Neural Network (CNN) Architecture for Multi-Classificationn

Recently, CNN’s models generated significant outcomes in different areas, such as NLP, image classification, and diagnosis systems [34]. In contrast to MLPs, CNN reduces the number of neurons and parameters, which results in lower complexity and faster adaptation. The CNN model has significant applications in medical image classification [34]. Here, we discuss the fundamental structure of the CNN model. The CNN is a type of Feed-Forward Neural Network (FFNN) and a DL model. Convolution operations can capture translation invariance, which means that the filter is independent of position, which reduces the number of parameters. The CNN model has three kinds of layers, such as Convolutional, Pooling, and fully connected layer. These three kinds of layers are necessary for performing functions of dimensionality reduction, feature extractors, and classification. During the convolution operation of the forward pass, the filter is slid onto the input volume and computes the activation map, which computes the point-wise output of each value and adds them to achieve the activation of that point. The sliding filter is deployed by convolution, and as a linear operator, it can be expressed as a dot product for fast deployment. Let us consider *x* is the input, and *w* is the kernel function, the convolution process (x∗w)(a) on time index *t* can be mathematically expressed in Equation (Equation 1).
(1)(x∗w)(a)=∫x(t)w(a−t)da
where *a* is in $R^{n}$ for any n≥1. While Parameter *t* is discrete. In this case, the discrete convolution can be expressed as in Equation (Equation 2):(2)$$\left(x*w\right)\left(a\right)=\sum_{a}x\left(t\right)w\left(t-a\right)$$

However, usually use 2 or 3-dimensional convolutions in CNN model. In this work, we used two dimensional convolutions CNN model for our multi-classification problem. In case of two-dimensional image *I* as input, *K* is a two dimensional kernel and the convolution can be mathematically expressed as in Equation (Equation 3):(3)$$\left(I*K\right)\left(i,j\right)=\sum_{m}\sum_{n}I\left(m,n\right)K\left(i-m,j-n\right)$$

Additionally, to gain non-linearities, two activation functions can be used, such as ReLU and Softmax. In Equation (Equation 4), the activation function ReLU expressed:(4)ReLU(x)=max(0,x)x∈R
the gradient of ReLU(x)=1 for x>0 and $ReLU^{-}\left(x\right)=0$ for x<0. The ReLU convergence capability is good then sigmoid non-linearities. The second activation function is softmax, which is expressed mathematically in Equation (Equation 5). The softmax non-linearity activation function is suitable when the output needs to be included in more than two classes.
(5)$$Softmax\left(x_{i}\right)=\frac{\exp(x_{i})}{\sum_{j}\exp\left(x_{j}\right)}.$$

The CNN model pooling layers are utilized to output a statistics summary of its inputs and resize the output shape without losing necessary information. There are different type of pooling, and we use maximum pooling layer which generates the maximum values in individually rectangular neighborhood of individual point (*i*, *j*) for 2D data of each input feature, respectively. A fully connected layer FC is last layer with *n* and *m* input, and output sizes are described below. The output layer parameters are expressed as a weight matrix $W\in M_{m,n}$. Where *m* rows, *n* columns, and a bias vector $b\in R^{m}$. Assumed an input vector $x\in R^{n}$, the fully connected layer FC output with activation function *f* is expressed mathematically in Equation (Equation 6) as:(6)$$FC\left(x\right):=f\left(Wx+b\right)\in R^{m}$$
in Equation (Equation 6), Wx is the product of the matrix, while the function *f* is applied component wise. The fully connected layers are utilized for problems of classification. The fully connected layers FC of CNN model are generally attached at the top. For this, the CNN output is flattened and showed as a single vector. In our proposed 2D CNN model, there are three 2D convolution layers with each layer have an activation layer and max-pooling layer and FC is last layer. Furthermore, we use Stochastic Gradient Descent (SGD) Optimization algorithm for our model optimization. The structure of our CNN model is given in Table 1.

#### 2.2.2. Transfer Learning to Improve 2DCNN Model Predictive Performance

To improve the 2DCNN model predictive capability, we employed transferred learning ResNet-50 model. The transfer learning (TL) techniques widely used in image classification tasks [20], COVID-19 sub-type recognition [35] and medical images filtering [36]. In this study, we incorporated the transfer learning ResNet-50 CNN pre-trained model to enhance the predictive performance of the proposed 2DCNN model. The ResNet-50 pre-train model is trained on imageNet data set and transferred the weights of the trained parameters to our 2DCNN model, and fine-tuned the model using the chest X-ray images for the final classification of the 2DCNN model.

The structure of ResNet-50 have 5 steps and each step with convolution, and identity block. In each block of convolution there are 3 layers of convolution, also three layers of convolution in each identity block. Furthermore, ResNets-50 is a variant of ResNet model, which has 48 Convolution layers along with 1 max-pool and 1 average pool layer. The ResNet-50 model has more than 74,917,380 trainable parameters. The architecture of ResNet-50 is given in Figure 1.

#### 2.2.3. Cross Validation Criteria

The holdout cross-validation mechanism is used for model training and validation [5,8]. In this study chest X-ray images data sets were divided into 70% and 30% for training and teasing of the model for all experiments.

#### 2.2.4. Model Assessment Criteria

In this work, important assessment measures [7] are used to evaluate the proposed method, which are expressed mathematically in Equations (Equation 7)–(Equation 12), respectively.
(7)$$Accuracy\left(Ac\right)=\frac{\left(TP+TN\right)}{\left(TP+TN+FP+FN\right)}\times 100$$
(8)$$Recall/Sensitivity\left(Re/Sn\right)=\frac{TP}{\left(TP+FN\right)}\times 100$$
(9)$$Specificity\left(Sp\right)=\frac{TN}{\left(TN+FP\right)}\times 100$$
(10)$$Precision\left(pr\right)=\frac{TP}{\left(TP+FP\right)}\times 100$$
(11)$$F1-score\left(F1S\right)=2\times \frac{Pr\times Re}{\left(pr+Re\right)}\times 100$$
(12)$$Matthewscorrelationcoefficient\left(MCC\right)=\frac{T_{1}}{\sqrt{T_{2}\times T_{3}\times T_{4}\times T_{5}}}x\times 100$$
where, $T_{1}=\left(TP\times TN-FP\times FN\right)$, $T_{2}=\left(TP+FP\right)$, $T_{3}=\left(TP+FN\right)$, $T_{4}=\left(TN+FP\right)$, and $T_{5}=\left(TN+FN\right)$

AUC: The AUC demonstrated the model ROC and large AUC value show good predictive results of the model.

#### 2.2.5. Proposed Integrated (ResNet-50+2DCNN) Multi Classification (R2DCNNMC) Model for COVID-19 Diagnosis

We have designed the 2DCNN model for COVID-19 detection employing chest X-ray images data. To improve the predictive performance of the 2DCNN model, we have used techniques of data augmentation and transfer learning (TL). We have used transfer learning pre-trained CNN architecture ResNet-50 [37]. The imagesNet data set has been employed for pre-trained of ResNet-50, and the generated weights (trained parameters) of this model are transferred for the training of our 2DCNN model. Chest X-ray data set is utilized for fine-tuning of the 2DCNN model and for final multi-classification of the model. Thus, an integrated (ResNet-50+2DCNN) multi-classification (R2DCNNMC) model for COVID-19 diagnosis is proposed.

A hold-out cross validation (CV) mechanism is used in the proposed R2DCNNMC model, with 70% of the model being trained and 30% being tested. The integration of transfer learning greatly enhanced the predictive performance of the 2DCNN model. The performance of the proposed R2DCNNMC model has been evaluated using evaluation metrics. The pseudo code for our model R2DCNNMC is given in Algorithm 1, and a flow chart is shown in Figure 2.
**Algorithm 1:** Proposed R2DCNNMC model for COVID-19 diagnosis.
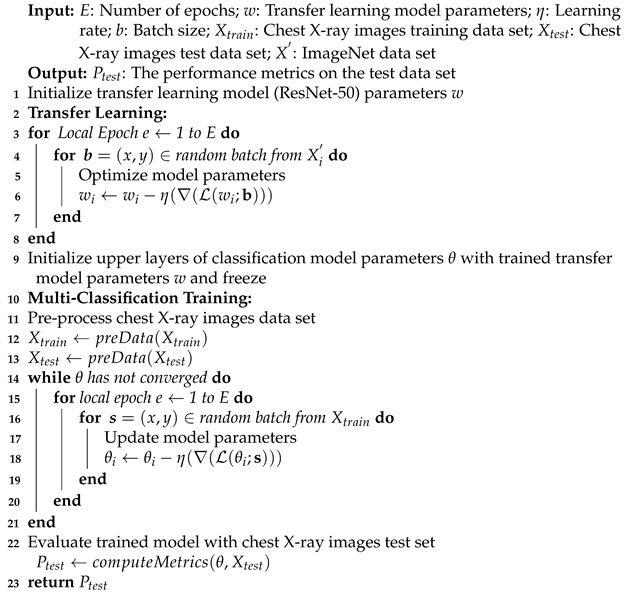


## 3. Experiments and Discussion

### 3.1. Experimental Setup

For implementation of our proposed R2DCNNMC model we have performed various experiments. For model validation two chest X-ray data sets have utilized and hold-out cross validation technique is used for model training and validation. Model assessment measures have computed for model evaluation. In addition Stochastic Gradient Descent (SGD) Optimization algorithm has been used for proposed model optimization. Others parameters such as learning rate α (SGD) = 0.0001, epochs = 120, batch size = 100, Mini-batch size = 9, outer activation function = Softmax and inner activation function = ReLU have been used in all experiments. In Table 2, the the proposed R2DCNNMC model parameters are defined accordingly. The hardware setup for all experiments we used a laptop with Intel Core i5, 64 GB RAM, and GPU. Python v3.7 is used for simulations and the proposed model is developed in Keras framework v2.2.4 and Tensor flow v1.12 as the back end. All experiments repeated many times for producing stable results.

### 3.2. Results and Analysis

#### 3.2.1. Pre-Processing of Data

Two data set are used in this research for the evaluation of the proposed R2DCNNMC model. Before applying these data sets in the model we need to perform so pre-processing operations on both data sets that model suitable trained for effective performance. The COVID-19-Radiography-Dataset (CRD) is a data set of chest X-ray images for COVID-19 positive cases along with Normal, Viral Pneumonia, and Lung Opacity. This data set included 3616 COVID-19 positive cases, 10,192 Normal, 6012 Lung Opacity, and 1345 Viral Pneumonia images. The total images in the data set are 21,165.

To increase the data set size for effective training of the 2DCNN model we have used the data augmentation technique to augment the original dataset by using random transformation (rotation). All the images have been rotated with an angle of 45 degrees along the X-axis and added these augmented images to the original data set. Thus, the total images in new data are 42,330. The data augmentation technique has also used on the second chest X-ray (Covid-19 and Pneumonia) (CXI) data set. This data set has chest X-ray 6432 images, which belong to three classes (COVID19, Normal, PNEUMONIA). The data set contain 576 COVID19, 1583 Normal, and 4273 PNEUMONIA images, respectively.

After data augmentation, the images in the new dataset are 12,864. The proposed model has trained on original and augmented data sets, respectively, for all experiments. The holdout cross-validation method has used for the training and validation process in the proposed model because the data sets are now large enough so it will not make computational complexity problems and model will fit exactly and will generate high performance. The images of CRD and CXI datasets are shown in Figure 3 and Figure 4.

#### 3.2.2. 2DCNN Model Performance Evaluation on Original and Augmented Date Sets

The predictive output of the 2DCNN model has been checked on two chest X-ray original and augmented data sets. The 2DCNN model has been trained with these two types of data sets along with other necessary hyperparameters. The SGD algorithm of optimization with a Learning Rate (LR) of 0.0001 is used in the model for model optimization. The number of epoch and batch sizes were 100 and 120, respectively, for all experiments. The results are reported in Table 3.

Table 3 is present the performance of 2DCNN model on original and augmented COVID-19-Radiography (CRD) chest X-ray data sets. According to Table 3, the 2DCNN model on original COVID-19-Radiography (CRD) chest X-ray data set has gained 95.20% Accuracy, 97.00% Specificity, 80.25% Sensitivity/Recall, 92.40% Precision, 93.00% MCC, 95.09% F1-score and 96.00% AUC, respectively. On the other side, the 2DCNN model on augmented COVID-19-Radiography (CRD) chest X-ray data obtained high performance as compared to the performance on the original data set. The 2DCNN model has achieved 96.00% Accuracy, 96.45% Specificity, 97.00% Sensitivity/Recall, 97.43% Precision, 96.33% MCC, 96.52% F1-score and 97.23% AUC, respectively, on augmented data set (CRD). The accuracy of the 2DCNN model has increased from 95.20% to 96.45% when the model trained with augmented COVID-19-Radiography (CRD) chest X-ray data set. Similarly, the AUC value of the 2DCNN model has increased from 96.00% to 97.23%. The other evaluation metrics values also improved with the data augmentation process.

The 2DCNN model performance has also been evaluated by using chest X-ray COVID-19, and Pneumonia (CXI) data set. Table 3 shows that the 2DCNN model has obtained 97.02% Accuracy, 98.00% Specificity, 99.25% Sensitivity/Recall, 100.00% Precision, 99.26% MCC, 97.00% F1-score and 99.00% AUC, respectively.

While with augmented chest X-ray COVID-19 and Pneumonia (CXI) data set the 2DCNN model has achieved 97.65% Accuracy, 99.10% Specificity, 97.86% Sensitivity/Recall, 99.80% Precision, 99.87% MCC, 97.73% F1-score and 99.23% AUC, respectively. Due to the data augmentation, the training of the 2DCNN was effectively performed and ultimately has increased model predictive performance. With the data augmentation process the model increased Accuracy from 97.02% to 97.65%, which demonstrates that the model predictive capability increased with data augmentation. Similarly, the MCC value has increased from 99.26% to 99.87%, and the AUC value has also improved from 99.00% to 99.23%.

#### 3.2.3. ResNet-50 Model Performance Evaluation on Original and Augmented Date Sets

The ResNet-50 model performance has been evaluated on two chest X-ray original and augmented data sets. The ResNet-50 transfer learning CNN model trained with two types of data sets along with other required hyperparameters. The SGD optimization algorithm with LR of 0.0001 is used in the model for model optimization. The number of epoch and batch sizes were 100 and 120, respectively, for all experiments. To evaluate model performance different evaluation metrics have computed and reported in Table 4.

Table 4 shows that the ResNet-50 model on original CRD data set has obtained 94.03% Accuracy, 96.32% Specificity, 83.25% Sensitivity/Recall, 97.10% Precision, 93.50% MCC, 95.09% F1-score and 94.20% AUC, respectively. On the other side, the ResNet-50 model on augmented CRD data set obtained high performance as compared to the performance on the original data set. The ResNet-50 model has achieved 95.20% Accuracy, 97.00% Specificity, 99.00% Sensitivity/Recall, 88.21% Precision, 96.12% MCC, 97.34% F1-score and 95.00% AUC, respectively, on augmented data set (CRD).

The accuracy of the ResNet-50 model has increased from 94.03% to 95.20% when the model trained with augmented CRD chest X-ray data set. Similarly, the AUC value of the ResNet-50 model has increased from 94.20% to 95.00%. The other evaluation metrics values also improved with the data augmentation process.

The ResNet-50 model performance has also been evaluated by using chest X-ray COVID-19 and Pneumonia (CXI) data set. According to Table 4 the ResNet-50 model has obtained 92.34% Accuracy, 94.46% Specificity, 97.67% Sensitivity/Recall, 93.00% Precision, 94.98% MCC, 93.00% F1-score and 92.10% AUC, respectively.

While with augmented CXI data set the ResNet-50 model has achieved 94.87% Accuracy, 95.98% Specificity, 95.51% Sensitivity/Recall, 93.00% Precision, 95.24% MCC, 95.00% F1-score and 93.19% AUC, respectively. Due to the data augmentation, the training of the ResNet-50 was effectively performed and ultimately has increased model predictive performance. With the data augmentation process the model increased accuracy from 92.34% to 94.87%, which demonstrates that the model predictive capability increased with data augmentation.

#### 3.2.4. R2DCNNMC Performance Evaluation on Original and Augmented Data Sets

The performance of the R2DCNNMC model has been evaluated on two chest X-ray original and augmented data sets. R2DCNNMC model has been trained with these two types of data sets along with other required hyperparameters. The SGD optimization algorithm with a LR of 0.0001 is used for model optimization. The number of epoch and batch sizes were 100 and 120, respectively, for all experiments. For training and validation of the model, 70% and 30% data are used. To evaluate model performance, different assessment measures have been computed and reported in Table 5.

The performance of the R2DCNNMC model on original and augmented COVID-19-Radiography (CRD) chest X-ray data sets is reported in Table 5. According to Table 5, the R2DCNNMC model on the original COVID-19-Radiography (CRD) chest X-ray data set has achieved 97.66% Accuracy, 99.00% Specificity, 89.18% Sensitivity/Recall, 99.10% Precision, 99.30% MCC, 98.00% F1-score and 97.03% AUC, respectively. On the other side, the R2DCNNMC model on augmented CRD data obtained high performance as compared to the performance on the original data set. The R2DCNNMC model has achieved 98.12% Accuracy, 99.28% Specificity, 93.00% Sensitivity/Recall, 99.56% Precision, 99.70% MCC, 98.23% F1-score and 98.60% AUC, respectively, on augmented CRD data set. The accuracy of the R2DCNNMC model has increased from 97.66% to 98.12% when the model trained with augmented CRD data set. Similarly, the AUC value of the 2DCNN model has increased from 97.03% to 98.60%. The other evaluation metrics values also improved with the data augmentation process.

The R2DCNNMC model performance has also been checked by using chest X-ray Covid-19 and Pneumonia (CXI) data set. According to Table 5, the R2DCNNMC model have obtained 98.17% Accuracy, 100.00% Specificity, 96.25% Sensitivity/Recall, 99.24% Precision, 99.70% MCC, 99.46% F1-score and 99.23% AUC, respectively. While with augmented chest X-ray Covid-19 and Pneumonia (CXI) data set the R2DCNNMC model has achieved 99.45% Accuracy, 99.63% Specificity, 96.99% Sensitivity/Recall, 100.00% Precision, 99.83% MCC, 99.78% F1-score, and 99.90% AUC, respectively. Due to the data augmentation the training of the R2DCNNMC effectively performed and ultimately has increased model predictive performance. With the data augmentation process the model increased Accuracy from 98.17% to 99.45%, which demonstrates that the model predictive capability increased with data augmentation. Similarly, the MCC value has increased from 99.70% to 99.83% and the AUC value has also improved from 99.23% to 99.90%.

#### 3.2.5. Proposed R2DCNNMC Model Performance Comparison with Baseline Methods

In Table 6, we compared the proposed R2DCNNMC model performance in terms of accuracy with baseline methods. Table 6 shows that the proposed model R2DCNNMC achieved 98.12% accuracy with data set CRD, which is higher than baseline methods. Similarly, the proposed model R2DCNNMC achieved 99.45% accuracy with data set CXI, which is higher than baseline models. The excellent predictive performance of the proposed model demonstrated that it correctly detected COVID-19 and that it can be easily deployed in E-health care for COVID-19 diagnosis.

#### 3.2.6. Discussion

COVID-19 is rapidly spreading, and many people are suffering and dying as a result of this global pandemic. Accurate and timely diagnosis is a significant medical challenge for effective COVID-19 control and treatment. Various techniques are used to control and diagnose this disease. Soft computing-based COVID-19 diagnosis methods are widely used, and numerous AI-based methods have been proposed by various researchers. However, these methods continue to suffer from a lack of accuracy in diagnosing COVID-19 patients.

The COVID-19 disease has a significant impact on the human respiratory system, and the lungs lose functionality quickly. Thus, using chest X-ray images to diagnose COVID-19 patients is an appropriate method that clinical professionals typically use. However, due to human error, medical doctors’ interpretation of chest X-ray images to diagnose COVID-19 is insufficiently accurate. As a result, AI-based interpretation methods for distinguishing between normal and COVID-19 patient chest X-ray images are more effective.

The deep learning techniques based COVID-19 detection method from chest X-ray images is significantly important for the accurate diagnosis of COVID-19. The CNN model has significant applications in medical image classification [34]. The CNN model extracts more deep features from images data, and these features can help in the final classification.

To tackle the accurate diagnosis problem of COVID-19 in this research study, we have proposed a model for COVID-19 diagnosis employed CNN, data augmentation, and transfer learning techniques. The CNN model is used for deep features extraction and classification. Data augmentation and transfer learning techniques are used to improve the predictive capability of the CNN model. Two COVID-19 chest X-ray images data sets are used for validation of the proposed model. These data sets are not insufficient for effective training of the model. Hence, we have used the data augmentation [47] technique to increased the size of these data sets to train the model effectively and achieve excellent performance. The experimental results show that the proposed model obtained high performance on both original and augmented data sets as compared to baseline methods. The major finding of this study are as follows:

Firstly, the accuracy of the 2DCNN model has increased from 95.20% to 96.45% when the model trained with augmented CRD data set. Similarly, the AUC value of the 2DCNN model has increased from 96.00% to 97.23%. In the 2DCNN model with augmented CXI data set, the accuracy improved from 97.02% to 97.65% and the MCC value increased from 9.26% to 99.87%, while the AUC value also improved from 99.00% to 99.23%. Thus, these results demonstrated that the model predictive capability increased with data augmentation.

Secondly, transfer learning techniques incorporated with the 2DCNN model and with CRD and CXI data sets the model increased accuracy from 97.66% to 98.12% and 98.17% to 99.45%, respectively.

Thirdly, the proposed model (R2DCNNMC) obtained 98.12% classification accuracy on CRD data set and 99.45% classification on CXI data set as compared to baseline methods. Due to higher predictive performance of the proposed model, we recommend it for accurate diagnosis of COVID-19 in E-healthcare.

## 4. Conclusions

Deep learning algorithms, particularly convolutional neural networks, are commonly used to analyze medical image data. The accurate diagnosis of COVID-19 is a critical issue, and a new accurate diagnosis method is significantly needed to address it. Hence to diagnosis COVID-19 accurately, we have proposed a R2DCNNMC model, which is based on deep and transfer learning. In the proposed model designing we have used 2DCNN model for deep features extraction, and classification of chest X-ray images data for recognition of COVID-19. Two data sets have utilized for the validation of the proposed model. Furthermore, data augmentation techniques have been used for increasing data sets size for effective training of the proposed model. In addition cross-validation and model assessment measures have been computed for model evaluation.

The experimental results demonstrated that the proposed R2DCNNMC diagnosis model has been obtained very high performance and obtained 98.12% classification accuracy on CRD data set and 99.45% classification on CXI data set as compared to baseline methods. We recommend the proposed method for effective COVID-19 identification in E-healthcare due to its high predictive performance. In the future, we will use advanced models of transfer learning, federated learning, and deep learning, as well as other types of data sets, to diagnose COVID-19.

## Figures and Tables

**Figure 1 sensors-21-08219-f001:**
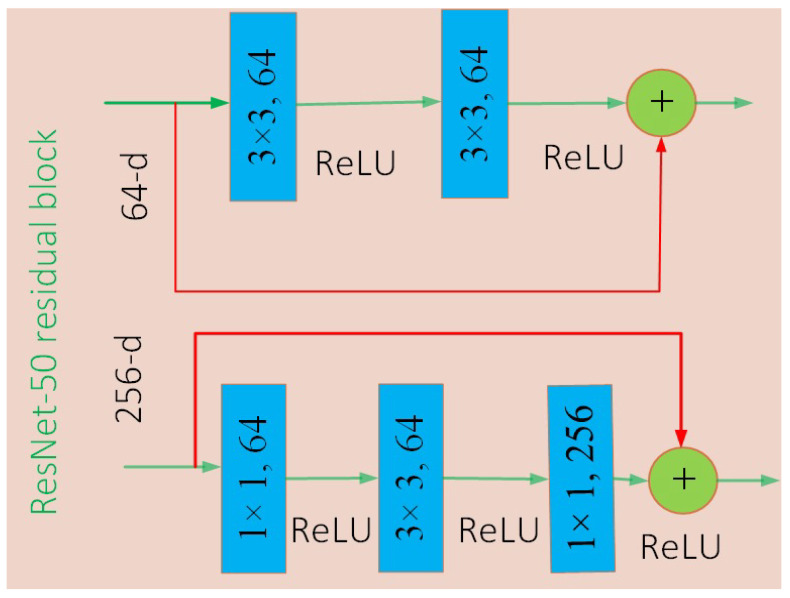
ResNet-50 architecture.

**Figure 2 sensors-21-08219-f002:**
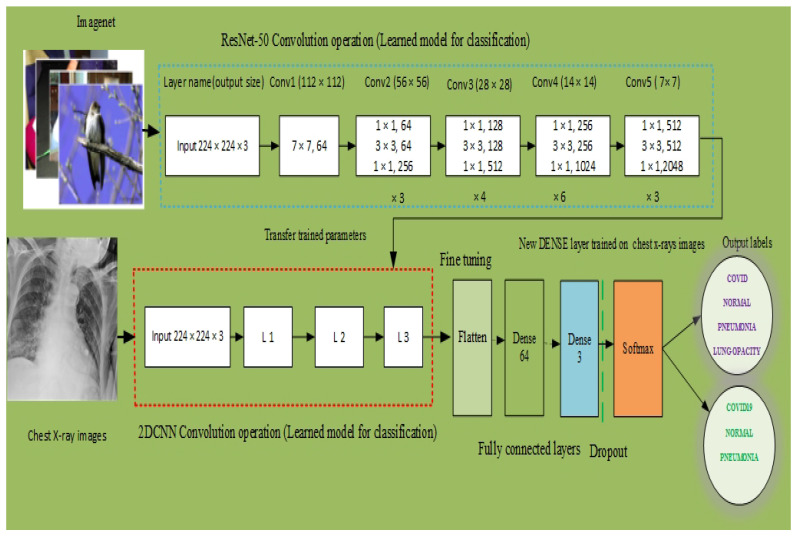
Proposed R2DCNNMC model for COVID-19 diagnosis.

**Figure 3 sensors-21-08219-f003:**
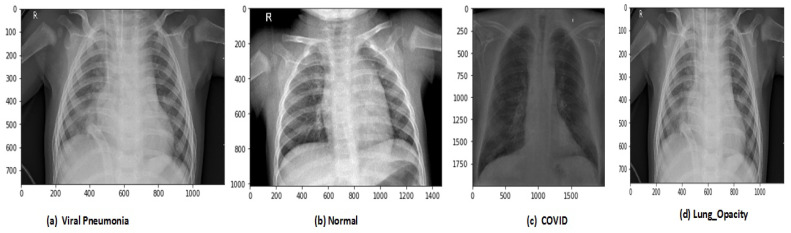
Types of chest X-ray images in CRD data set.

**Figure 4 sensors-21-08219-f004:**
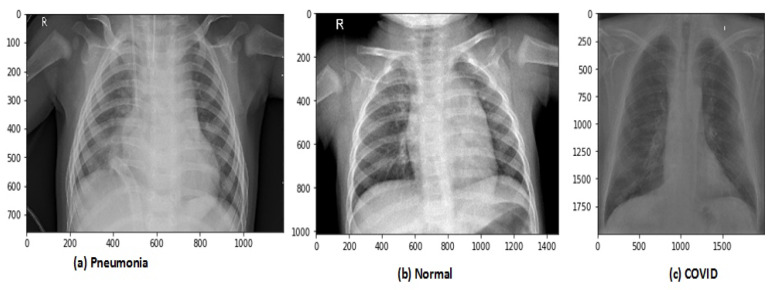
Types of chest X-ray images in CXI data set.

**Table 1 sensors-21-08219-t001:** 2DCNN model structure for multi-classification.

Number	Layer (Name)
1	Conv2D (64, (7, 2))
2	Activation (‘ReLU’)
3	MaxPool2D (pool-size = (2, 2))
4	Conv2D (64, (3, 3)
5	Activation (‘ReLU’)
6	MaxPool2D (pool-size = (2, 2))
7	Conv2D (64, (3, 3))
8	Activation (‘ReLU’)
9	MaxPool2D (pool-size = (2, 2))
10	Flatten ()
11	Dense (64)
12	Activation (‘ReLU’)
13	Dropout (0.5)
14	Dense (3)
15	Activation (‘Softmax’)

**Table 2 sensors-21-08219-t002:** R2DCNNMC model parameters.

Parameters	Value
Optimizer	SGD
Learning rate α	0.0001
Number of epoch	100
Bach size	100
Mini-batch size	9
Training data set	70%
Validation data set	30%

**Table 3 sensors-21-08219-t003:** 2DCNN model performance evaluation on original and augmented CRD and CXI data sets.

Data Set		Parameters		Assessment Measures						
		**Optimizer**	**LR**	**Ac (%)**	**Sp (%)**	**Sn/Re (%)**	**Pr (%)**	**MCC (%)**	**F1S (%)**	**AUC (%)**
CDR	original	SGD	0.0001	95.20	97.00	80.25	92.40	93.00	95.09	96.00
	augmented	-	-	96.00	96.45	97.00	97.43	96.33	96.52	97.23
CXI	original	-	-	97.02	98.00	99.25	100.00	99.26	97.00	99.00
	augmented	-	-	97.65	99.10	97.86	99.80	99.87	97.73	99.23

**Table 4 sensors-21-08219-t004:** ResNet-50 model performance evaluation on original and augmented CRD and CXI data sets.

Data Set		Parameters		Assessment Measures						
		**Optimizer**	**LR**	**Ac (%)**	**Sp (%)**	**Sn/Re (%)**	**Pr (%)**	**MCC (%)**	**F1S (%)**	**AUC (%)**
CDR	original	SGD	0.0001	94.03	96.32	83.25	97.10	93.50	95.00	94.20
	augmented	-	-	95.20	97.00	99.00	88.21	96.12	97.34	95.00
CXI	original	-	-	92.34	94.46	97.67	98.00	94.98	93.00	92.10
	augmented	-	-	94.87	95.98	95.51	93.00	95.24	95.00	93.19

**Table 5 sensors-21-08219-t005:** R2DCNNMC model performance evaluation on original and augmented CRD and CXI data sets.

Data Set		Parameters		Assessment Measures						
		**Optimizer**	**LR**	**Ac (%)**	**Sp (%)**	**Sn/Re (%)**	**Pr (%)**	**MCC (%)**	**F1S (%)**	**AUC (%)**
CRD	original	SGD	0.0001	97.66	99.00	89.18	99.10	99.30	98.00	97.03
	augmented	-	-	98.12	99.28	93.00	99.56	99.70	98.23	98.60
CXI	original	-	-	98.17	100.00	96.25	99.24	99.70	99.46	99.23
	augmented	-	-	99.45	99.63	96.99	100.00	99.83	99.78	99.90

**Table 6 sensors-21-08219-t006:** R2DCNNMC model accuracy comparison with baseline methods.

Method	Accuracy (%)	Reference
ResNet + SVM	95.38	[38]
GAN + Resnet18	99	[32]
VGG-16+ CNN	91.24	[39]
TLRV1	94.4	[40]
DTL	95.72	[41]
ResNet-50	96.23	[42]
COVID-Net-TM	92.4	[33]
DRE-Net	86	[43]
COVIDx-Net	90	[44]
TM	93.3	[33]
TL	93	[45]
COVID-Net CT-2	98.1	[30]
DarkCovidNet	90.8	[32]
ResNet50	90	[46]
VGG19 + CNN	98.05	[35]
VGG-19	98.87	[29]
Proposed method (R2DCNNMC)	98.12	2021
Proposed method (R2DCNNMC)	99.45	2021

## Data Availability

The data sets used in this study available on public repositories.

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
