# Peer review of "Diagnostic Approach for Accurate Diagnosis of COVID-19 Employing Deep Learning and Transfer Learning Techniques through Chest X-ray Images Clinical Data in E-Healthcare"

_sensors, 2021, doi:10.3390/s21248219_

Round 1

Reviewer 1 Report

This paper developed a deep learning algorithm, R2DCNNMC, to diagnose COVID-19 lesions on chest X-ray images. The proposed algorithm was based on the 2-dimensional Convolution Neural Networks (2DCNN) model for extraction of deep features on chest x-ray for initial classification. Then, the ResNet-50, which was pre-trained on ImageNet data set, was used to transfer the weights of the trained parameters to the 2DCNN model, and fine-tune the model using the chest X-ray images for the final classification. Two x-ray datasets were used: CRD and CXI. The results showed that the proposed R2DCNNMC using 2DCNN+ResNet50 obtained 98.12% classification accuracy on CRD data set, and 99.45% accuracy on CXI data set, which were higher as compared to the performance of the baseline models. Although the results showed higher accuracy, the improvement was marginal and very likely non-significant. Some methodology issues and the clinical value of this paper need to be clarified. Major comments below:

  1. The paper needs to be professionally edited, e.g., by a scientific agency. Although the technical methods and results were easy to follow, the general description part was poorly written. There were numerous grammar errors.
  2. The paper was aiming for the diagnosis of COVID-19 lesions using chest x-ray. While this was an important method during the early outbreak in 2020, now there are well-established and reliable PCR and rapid antigen tests. The clinical purpose of diagnosis based on X-ray needs to be carefully revised according to the real value of this work at the present time, not in the past.
  3. Two datasets were used: CRD and CXI, both containing chest x-ray images with confirmed COVID-19 lesions, viral-related pneumonia, and normal images. They presented a great opportunity for testing the model trained on one dataset in an independent dataset. But, the authors used two datasets separately, based on 70% training and 30% hold-out testing in each dataset.
  4. The natural images and chest x-ray images in Figure 2 were distorted, which raised some doubts about their image processing methods. Please make sure the x-ray images were open correctly and processed using a standardized method.
  5. Please show some examples of the chest x-ray images with different diagnoses, and also please illustrate how the images were processed. For example, how large is the input box, the entire x-ray image, or an area covering the lesion? If the latter, how was the lesion ROI segmented, and how was the input box size determined?
  6. Figures 3-13 were used to graphically show the numbers already reported in the tables. They are not informative and should be removed. Instead, please show important information that needs to be illustrated graphically, e.g. images as suggested in issue 5).
  7. The higher accuracy obtained using the augmented data was highly anticipated, and not worth reporting. If the authors prefer, can specify that all results were obtained using the augmented datasets.
  8. The tables should be combined so the readers can easily compare the performance of different models.
  9. The difference in the improved accuracy seems to be very small, and not likely to reach a statistical significance level. Did the authors perform the statistical comparison between different methods?
  10. The discussion was simply repeating all numbers reported in the Results section, not informative. It should be rewritten to discuss the significant value of this work.

Author Response

Original Manuscript: Manuscript ID: sensors-1464227

Article Title: “Diagnostic Approach for Accurate Diagnosis of COVID-19 Employing Deep learning and Transfer learning Techniques Through Chest X-ray Images Clinical Data in E-healthcare".

To: Sensors

Re: Response to Editor, Associate Editor's and Reviewers

We would like to thank you all for giving us useful comments for further improvement of the manuscript. We have incorporated all the reviewer's comments in our updated manuscript successfully.

Best regards,

et al.

Response to reviewer's comments

Reviewer #1: his paper developed a deep learning algorithm, R2DCNNMC, to diagnose COVID-19 lesions on chest X-ray images. The proposed algorithm was based on the 2-dimensional Convolution Neural Networks (2DCNN) model for extraction of deep features on chest x-ray for initial classification. Then, the ResNet-50, which was pre-trained on ImageNet data set, was used to transfer the weights of the trained parameters to the 2DCNN model, and fine-tune the model using the chest X-ray images for the final classification. Two x-ray datasets were used: CRD and CXI. The results showed that the proposed R2DCNNMC using 2DCNN+ResNet50 obtained 98.12% classification accuracy on CRD data set, and 99.45% accuracy on CXI data set, which were higher as compared to the performance of the baseline models. Although the results showed higher accuracy, the improvement was marginal and very likely non-significant. Some methodology issues and the clinical value of this paper need to be clarified. Major comments below:

Author's response:  Thank you for your comments. We have incorporated your comments in the revised version.

Concern#1: The paper needs to be professionally edited, e.g., by a scientific agency. Although the technical methods and results were easy to follow, the general description part was poorly written. There were numerous grammar errors.

Author's response:   Thank you, we have rectified all grammatical errors and proofread the manuscript by English langue experts.

Concern#2: The paper was aimed at the diagnosis of COVID-19 lesions using chest x-rays. While this was an important method during the early outbreak in 2020, now there are well-established and reliable PCR and rapid antigen tests. The clinical purpose of diagnosis based on X-ray needs to be carefully revised according to the real value of this work at present, not in the past.

Author's response:  Thank you, in covid-19 disease majorly affected the respiratory system of the human being and the performance of lungs leaving work very quickly. To diagnosis COVID-19 from chest x-ray images using deep learning techniques based automatic diagnosis approach is more effective for accurate and on-time detection of COVID-19 for early treatment and recovery. So to the best of our knowledge, the chest x-ray based diagnosis approach is significantly necessary for precise diagnosis of COVID-19 and is economically also reliable.

Concern#3: Two datasets were used: CRD and CXI, both containing chest x-ray images with confirmed COVID-19 lesions, viral-related pneumonia, and normal images. They presented a great opportunity for testing the model trained on one dataset in an independent dataset. But, the authors used two datasets separately, based on 70% training and 30% hold-out testing in each dataset.

Author's response:  Thank you, the labels (output classes) in both data sets are different, so we used each data set separately for training and validation. So we do not use one data set for training and another for testing.

Concern#4:  The natural images and chest x-ray images in Figure 2 were distorted, which raised some doubts about their image processing methods. Please make sure the x-ray images were open correctly and processed using a standardized method.

Author's response:  Thank you, we redraw figure 2 for more visibility in the updated manuscript.

Concern#5:  Please show some examples of the chest x-ray images with different diagnoses, and also please illustrate how the images were processed. For example, how large is the input box, the entire x-ray image, or an area covering the lesion? If the latter, how was the lesion ROI segmented, and how was the input box size determined?

Author's response:  Thank you, in this work, we only increased the data set size by using data augmentation for practical model training. See section 3.2.1.

Concern#6. Figures 3-13 were used to graphically show the numbers already reported in the tables. They are not informative and should be removed. Instead, please show important information that needs to be illustrated graphically, e.g. images as suggested in issue 5).

Author's response:  Thank you, we have removed figures 3-13 from our updated manuscript.

Concern#7:  The higher accuracy obtained using the augmented data was highly anticipated, and not worth reporting. If the authors prefer, can specify that all results were obtained using the augmented datasets.

Author's response:   Thank you, data augmentation process is only incorporated in this study to increase the size of the data set to improve the model's predictive performance and the robustness of the proposed model. Due to this, the proposed model gained improved performance, as reported in the manuscript.

Concern#8: The tables should be combined so the readers can easily compare the performance of different models.

Author's response:   Thank you, we have combined tables in the updated manuscript.

Concern#9: The difference in the improved accuracy seems to be very small, and not likely to reach a statistical significance level. Did the authors perform the statistical comparison between different methods?

Author's response:  Thank you, the predictive accuracy of the proposed method is high as compared to baseline methods.

Concern#10: The discussion was simply repeating all numbers reported in the Results section, not informative. It should be rewritten to discuss the significant value of this work.

Author's response:   Thank you, we have revised the discussion according to the work presented in the manuscript.

Reviewer 2 Report

This paper proposes 2D Convolutional Neural Networks to diagnose COVID-19 using Chest X-Ray images.

The proposed method is not innovative, but the performance is spectacular.

First of all, the authors should explain more clearly why the proposed 2DCNN model has 3 Convolutional layers followed by two dense layers. That is, they should answer the following questions: why they adopt 3 layers for feature extraction, why the first two layers have 32 neurons, why they used SGD optimizer instead of Adam optimizer? Why they used a dropout of 0.5? Which initialization methods are used in each layer?

There are some typos that need to be corrected.

Page 1, line 2: “Currently” should be “currently.”

Page 3, line 95: “bellows” should be “follows.”

Page 4, line 126: “relu-(x)” should be “relu(x).”

Page 5, Table 1: Opening and closing parentheses are not paired. Please fix them.

Page 5, line 132: “Descend” should be “Descent.”

Page 6, Equation 11: “Precision” and “pr” are used interchangeably. Please change “Precision” to “pr” and add a multiplication sign (x) after the number 2 for clarification.

Page 6, line 166: “Diagnosis” should be “diagnosis.”

Author Response

Original Manuscript: Manuscript ID: sensors-1464227

Article Title: “Diagnostic Approach for Accurate Diagnosis of COVID-19 Employing Deep learning and Transfer learning Techniques Through Chest X-ray Images Clinical Data in E-healthcare".

To: Sensors

Re: Response to Editor, Associate Editor's and Reviewers

We would like to thank you all for giving us useful comments for further improvement of the manuscript. We have incorporated all the reviewer's comments in our updated manuscript successfully.

Best regards,

et al.

Reviewer #2   

This paper proposes 2D Convolutional Neural Networks to diagnose COVID-19 using Chest X-Ray images.

Author's response: Thank you, for your comments and we have incorporated your comments in an updated version of the manuscript.  

Concern#1: First of all, the authors should explain more clearly why the proposed 2DCNN model has 3 Convolutional layers followed by two dense layers. That is, they should answer the following questions, 

Author's response:  Thank you. It is worth noting that there is no one-fit model for all deep learning tasks. Specifically, each deep learning task requires its model architecture definition that helps achieve a good performance regarding the accuracy and computational efficiency.  With the above assertion, upon several model configuration and ablation experiments, the simple but effective model configuration of 3 convolutional layers and two dense layers achieve a convincing performance for this current research problem, making us settle for this model configuration. Although other models with very deep layer configurations performed well, they suffered much overfitting irrespective of dropout, and their accuracies difference with our proposed method was negligible. Furthermore, this current work utilizes the technique of transfer learning where we leveraged the learned parameters (e.g., weight and biases) of the top/upper layers from an already pre-trained model. Because of the transfer learning technique, there is no need to train a very complex model from scratch but instead rely on simple model configuration.

Concern#2: why they adopt 3 layers for feature extraction, 

Author's response:  Thank you. We adopted the three layers for feature extraction based on the exact reasons for Concern#1: ablation studies and transfer learning.

Concern#3: why the first two layers have 32 neurons.

Author's response:  Thanks.  The first two layers of our architecture have 64  neurons instead of 32, with the first layer having a  7x7 kernel size and 2x2 stride size. We have corrected the typo in the updated manuscript's current version.  To use the weights of the first convolution layer of the pre-trained Resnet-50,  we set the in-channel neuron of our model to 64 (with the first layer having a  7x7 kernel size and 2x2 stride size) to fit the weight dimension. Specifically, we restored and froze the weights of the first convolutional layer of the pre-trained Resnet150 imagnet model to our 2DCNN architecture and trained the remaining layers of our model. All other layers of the proposed model were configured by several ablation experiments. From performed experiments, we observed that restoring only the weights of the topmost/upper layer (only the first convolutional layer) of the Resnet-50 model as a transferring learning technique for this current task achieved significant results, as showed the experimental results of this current work.

Concern#4:    why they used SGD optimizer instead of Adam optimizer?

Author's response:  Thank you, we have performed experiments with both optimizers, but the model's performance with the SGD optimizer is higher than that of the Adam optimizer. Furthermore, SGD has better-generalized capability than Adam. Hence we only report the performance of the model with SGD optimizer in our manuscript.

Concern#5:    Why they used a dropout of 0.5?

Author's response:  Thank you, the dropout is used to regularize the network to prevent the model from overfitting during the model training. In our model, we used dropout (p=0.5) before the output layer, and at 0.5, the model predictive accuracy was convincing compared to other experimented dropout values.

Concern#6:   Which initialization methods are used in each layer?

Author's response:  Thank you, for each layer, we used the glorot uniform initializer, also known as the Xavier uniform initializer, for the parameter initialization.

Concern#7:   There are some typos that need to be corrected.

Page 1, line 2: "Currently" should be "currently."

Page 3, line 95: "bellows" should be "follows."

Page 4, line 126: “relu-(x)” should be “relu(x).”

Page 5, Table 1: Opening and closing parentheses are not paired. Please fix them.

Page 5, line 132: "Descend" should be "Descent."

Page 6, Equation 11: "Precision" and "pr" are used interchangeably. Please change "Precision" to "pr" and add a multiplication sign (x) after the number 2 for clarification.

Page 6, line 166: "Diagnosis" should be "diagnosis."

Author's Response:  Thank you, we have rectified all errors from our updated manuscript.

Reviewer 3 Report

The main question addressed by the research is Chest X-ray images  data auto-classification to diagnose COVID-19.

The topic considered in many research work in last three years, the authors hypnosis’s that “The existing diagnosis methods of COVID-19 have 8 the problem of lack of accuracy to diagnosis.” Is not true. There is much research consider the detection of COVID-19 with high accuracy as:

 Loddo, A.; Pili, F.; Di Ruberto, C. Deep Learning for COVID-19 Diagnosis from CT Images. Appl. Sci. 2021, 11, 8227. https:// doi.org/10.3390/app11178227

This paper adds some different deep learning architecture models and their results.

Figures need to be improved.

Many Recent work have been made to xray for covid detection,  further investigation and comparison may be required.  Authors should Compare their work with more recent deep learning models for COVID-19 auto-detection.

Some new references from 2021 are missing as mentioned before.

Author Response

Original Manuscript: Manuscript ID: sensors-1464227

Article Title: “Diagnostic Approach for Accurate Diagnosis of COVID-19 Employing Deep learning and Transfer learning Techniques Through Chest X-ray Images Clinical Data in E-healthcare".

To: Sensors

Re: Response to Editor, Associate Editor's and Reviewers

We would like to thank you all for giving us useful comments for further improvement of the manuscript. We have incorporated all the reviewer's comments in our updated manuscript successfully.

Best regards,

et al.

Reviewer #3    Thank you for reviewing our manuscript.

Concern#1: Figures need to be improved.

Author's response:  Thank you, we have redrawn all figures professionally.

Concern#2: Many Recent work have been made to x-ray for COVID-19 detection; further investigation and comparison may be required.

Author's response:  Thank you, we have compared the proposed method with baseline methods in table 6 in the updated manuscript.

All the above explanations have been added to the manuscript. We hope that such explanations can meet the reviewer's comments.

Concern#3: Some new references from 2021 are missing as mentioned bellow: This paper adds some different deep learning architecture models and their results.

Loddo, A.; Pili, F.; Di Ruberto, C. Deep Learning for COVID-19 Diagnosis from CT Images. Appl. Sci. 2021, 11, 8227. https:// doi.org/10.3390/app11178227

Author's response: Thank you, we have added new published work in our updated manuscript and also included the mentioned published paper.

Round 2

Reviewer 1 Report

Some x-ray images in Figures 3 and 4 are still distorted. Please ask a radiologist or a medical professional familiar with chest x-ray to check and make sure the images are correctly presented.

Author Response

Respected reviewer,

Thank you for your comments we have updated figures 3 and 4 in the updated manuscript.
